# Why Should Diabetic Women Be Active?—The Role of Personality, Self-Esteem, Body-Esteem, and Imagery

**DOI:** 10.3390/healthcare12080857

**Published:** 2024-04-18

**Authors:** Dagmara Budnik-Przybylska, Malwina Fituch, Aleksandra Kowalewska

**Affiliations:** 1Sport Psychology Division, Institute of Psychology, Faculty of Social Science, University of Gdansk, 80-309 Gdansk, Poland; 2Mindfitness, 57-340 Duszniki Zdrój, Poland; mf.mindfitness@gmail.com; 3Institute of Psychology, Faculty of Social Science, University of Gdansk, 80-309 Gdansk, Poland; kowalewska.1997@gmail.com

**Keywords:** diabetes, personality, physical activity, self-esteem, body-esteem, imagery

## Abstract

Diabetes is one of the fastest spreading diseases in the 21st century. The aim of the study is twofold: (1) to find differences in personality traits, self-esteem, body-esteem, and imagery between healthy women and women with diabetes; (2) to verify whether there are differences in the analyzed factors among women with diabetes who engage in sports compared to those who do not. We used 3 questionnaires: Imagination in Sport—short form, Self-Esteem Scale (SES), and BFIS Personality, which were tested online. We found that women with diabetes were characterized by significantly higher neuroticism, lower extraversion, and higher conscientiousness (marginally significant). We also found that women with diabetes who practice sport rated their bodies more highly in terms of sexual attractiveness and made better use of imagined affirmations than women without diabetes who were not active. Our study provides new insights into diabetics in terms of women navigating the disease.

## 1. Introduction

Diabetes is among the fastest-spreading diseases in the 21st century. It affects an increasing number of adults and children. Currently, the number is close to 430 million people worldwide, including 3 million people in Poland—more women (6.1%) than men (5.1%) [1,2]. Moreover, an increasing number of women are inactive [3,4]. Physical activity seems to be a natural cure or medicine for diabetes [5]. With increased and regular physical activity, the risk of developing type 2 diabetes, as well as the risk of coronary heart disease and vascular complications in people with type 1 diabetes, is reduced [6]. Regular exercise may even prevent or delay type 2 diabetes development [7]. It aids not only in achieving complete blood glucose control but also, in combination with an appropriate diet, helps to manage the disease [8,9].

In a study conducted by Attanayake et al. [10], the research revealed that people with diabetes were engaging in less physical activity compared to those without diabetes, and this was consistent over time. The strongest predictors of physical activity for people with diabetes, whether or not they engaged in sport, were gender and BMI. People with diabetes were less active and had a higher BMI than people without diabetes. Other issues that may contribute to low activity among people with diabetes include physical or cognitive disability, depressive symptoms, and poor sleep [10].

People with type 2 diabetes often have problems with obesity, which is another reason why regular physical activity is recommended [8]. Obesity, particularly truncal obesity, is closely correlated with the prevalence of diabetes and cardiovascular disease [4]. Excess body fat results in reduced glucose tolerance and reduced insulin sensitivity, which in turn leads to insulin resistance and diabetes. Specific exercise (Pilates) with a moderate intensity may improve the functionality of older women as well as sharply reduce blood glucose levels—between 28 and 30% [11]—with diabetes type 2. Studies show that people feel better after doing physical activity [12]. Happiness hormones such as dopamine, serotonin, and endorphins are produced during and after physical activity [13]. They have a soothing effect on the nervous system. A short duration of any type of exercise seems to reduce the risk of all-cause mortality and serious adverse events in patients with either hypertension, type 2 diabetes, or cardiovascular diseases [14].

It has been proven in many studies that physical activity has a huge impact on self-esteem and self-worth [11,15]. It has also been shown that people who participate in sports have higher self-esteem than those who do not [16]. Improved motor skills and the acquisition of new skills support this. By engaging in sports activities, people gain a capacity for objective perception of themselves and a realistic assessment of their abilities, as the physical activity they do is meaningful to them and they approach it with a positive emotional attitude [11,15]. Cultivating self-esteem can foster resilience and capabilities, thereby enhancing prospects for a positive future outlook. Self-esteem serves as a quality that amplifies one’s self-efficacy and convictions, potentially influencing an individual’s capacity to attain their objectives and succeed in their professional endeavors [17]. In diabetics, psychological well-being and attitudes toward life can deteriorate [18]. Type 1 diabetes typically presents itself before the age of 15 [19]. Consequently, the child, and later the adolescent, confronts the psychological challenges of the condition, a burden not shared by their healthy peers. This experience does not escape the notice of the developing personality [20]. As a metabolic and chronic disease, poorly managed diabetes is associated with many complications and carries the risk of developing disorders in the physical as well as the psychological sphere. The more deteriorated the somatic state of a patient, the more diminished their quality of life becomes [21]. Quality of life is largely determined by self-esteem; therefore, higher self-esteem leads to an improved quality of life [22].

The level of self-esteem is strongly correlated with the subjective evaluation of one’s own body. For people with diabetes, it is not easy to maintain a normal body weight, which results in changes in self-esteem and evaluation of one’s body. Individuals with diabetes appear to exhibit lower levels of body self-esteem compared to the general population, and this is notably linked to the clinical and psychological aspects of diabetes progression. Moreover, these differences and associations vary based on gender [23]. Patients with a normal body weight also have positive self-esteem but are characterized by higher levels of self-criticism. Obese or overweight individuals are characterized by higher self-esteem in areas other than physical appearance [22]. Individuals who engage in physical activity enhance their self-esteem and self-confidence. Self-belief correlates with a positive evaluation of one’s capabilities, and vice versa—lack of self-confidence correlates with low self-esteem. Also, high self-esteem positively correlates with playing performance [24].

Diabetes is a disease that tends to be chronic. Continuous glucose monitoring facilitates the anticipation of future glucose concentration trends, aiding in informed decisions regarding diabetes management. Glucose concentration values are influenced by a range of physiological and metabolic factors, including physical activity and acute psychological stress, along with the effects of meals and insulin [25]. This is why people with diabetes need to watch their diet, constantly monitor their blood glucose levels, remember to take insulin or other medicine, and engage in regular physical activity. Diabetes is also associated with stressful situations and psychological burdens that can affect an individual’s personality, especially those at a young age [26].

Several studies have examined the personalities of people with diabetes and discovered that they exhibit certain traits that are distinct from those of healthy individuals [20,24,27,28,29,30,31]. However, the results of the studies vary. Some studies highlight high emotional instability along with a tendency towards explosiveness and irritability in children and adolescents compared to their peers [20]. In interpersonal interactions, individuals with diabetes may exhibit withdrawal and a propensity to become reliant and submissive to others. Additionally, it has been found that people with diabetes display a strong need for social contact [22].

One of the theories of personality, the Big Five personality traits, comprises five distinct traits used in the study of personality. The theory identifies five factors and ten aspects. These factors and values are as follows: openness to experience (imaginative/inquisitive vs. consistent/cautious) conscientiousness (efficient/organized vs. extraverted/careless), extraversion (sociable/energetic vs. solitary/reserved), agreeableness (friendly/compassionate vs. critical/judgmental), and neuroticism (sensitive/nervous vs. resilient/confident) [32]. People with diabetes are more conscientious, responsible, and persistent, with greater emotional maturity [24]. Ruszczyńska [27] also proved in her research that those patients are characterized by greater neuroticism, manifested by strong anxiety. In a study conducted by Wheeler et al. [28] on 28 patients with type I diabetes, it was revealed that neuroticism was negatively related, while conscientiousness was positively related to adherence to the overall diabetes regimen. Studies by Esmaeilinasab et al. [29] uncovered that extraversion and conscientiousness can help control blood sugar, whereas anxiety and neuroticism correlate negatively with glycemic control. Similarly, Szymańska [20] found that people with diabetes are less resistant to stress. The findings of the study by Mukherjee et al. [30] showed significantly higher neuroticism in all diabetics compared to healthy controls. One of the key hypotheses that explains this correlation is the psychological burden hypothesis [30]. It states that stress associated with the awareness of having diabetes or any other chronic disease or health complications, can lead to the development of depression or other negative psychological states [30].

Basinska and Ratajska [31] mentioned that people with autoimmune diseases, including diabetes, are characterized by common personality traits. They are calm, introverted, trustworthy, conscientious, adaptable to their environment, sensitive to criticism, stubborn, rigid, and reserved. When examining neuroticism and physical activity, there was no significant association found between the two. Overall, neuroticism appears to be negatively related to physical activity but has little effect [33].

In the cross-sectional study, Woon et al. [34] studied the Big Five Personality Traits and Quality of Life in Elderly Malaysian Patients with Diabetes Mellitus. The results of the study undoubtedly confirmed the hypothesis that personality traits may play an important role in moderating the impact of diabetes on patients’ lives, even among elderly diabetic patients who were likely to have lived with the disease for a long time.

Traits such as conscientiousness and agreeableness may act as protective factors for the ‘Quality of life’ variable in older patients with diabetes, while the trait of neuroticism may potentially be a risk factor for reduced ‘Quality of life’ in these patients [34].

Research suggests that high conscientiousness and low neuroticism are related to positive health behaviors. Specifically, higher conscientiousness and lower neuroticism were associated with more physical activity [35]. Enhancements in systemic, and potentially hepatic, insulin sensitivity resulting from physical activity may persist from 2 to 72 h. The decrease in blood glucose levels is closely correlated with the duration and intensity of physical activity [36].

People who practice sports more often use imagery as a mental training technique and present higher imagery abilities than novices or nonathletes [37,38]. Imagery is a multifaceted and broadly applicable concept that can be used as a general ability in an individual’s daily life, but it is often used for performance optimization, therapy, and personal development, or problem-solving [38,39,40,41,42,43]. Imagery involves the creation of multisensory mental depictions of actions (or objects and situations) in the absence of direct experience and corresponding sensory input [44,45]. It can be recalled from memory or be a new combination of stimuli [46].

Motor imagery is important for an athlete because, by refining and transforming it, she/he tries to perform the movement as she/he has imagined it. It is an externalized pattern through which the athlete can learn. With the help of movement imagery, the individual can control the movements being performed and analyze errors and imperfections to improve them [47].

The vividness of positive future imagery was significantly linked with optimism, regardless of socio-demographic variables and regular use of mental imagery. The ability to generate vivid mental imagery of positive future events may serve as an adaptable indicator of optimism. Enhancing positive future imagery could provide a cognitive target for innovative treatments aimed at fostering optimism, with implications for mental health as well as physical well-being [48]. Findings from the research of Odou and Vella-Brodrick [49] suggest that positive psychology interventions as well as practicing imagery can foster well-being and the subsequent benefits of well-being.

The aim of the study is twofold: (1) to find differences in personality traits, self-esteem, body-esteem, and imagery between healthy women and women with diabetes; and (2) to verify whether there are differences in the analyzed factors among women with diabetes who engage in sports compared to those who do not.

The study poses the following research questions:(1)Are there differences between healthy and diabetic women in the level of personality traits, self-esteem, body esteem, and ability to use imagery?(2)Are there differences between diabetic women participating in sport and not participating in sport in levels of personality traits, self-esteem, body esteem, and ability to use imagery?

The following hypotheses are put forward:Women with diabetes are more neurotic and conscientious than healthy women. According to Woon et al. [34], neuroticism is associated with a diminished subjective sense of diabetic management and heightened levels of anxiety, particularly among women with diabetes mellitus (DM). Enhancing self-efficacy in diabetes management could potentially aid individuals in coping with anxiety symptoms, especially among those exhibiting traits of neuroticism.Healthy women who participate in sports assess their bodies better in terms of physical fitness compared to healthy women who do not participate in sports. Moreover, they are characterized by higher imagery.

A study by Ziemianek et al. [50], in which the authors examined the self-esteem and body image of women and men who exercise at the gym, showed that physically active people rate their bodies above average. In a study by Budnik-Przybylska et al. [51], dancers showed higher imagery abilities as well as better ratings of their bodies in terms of physical fitness.

## 2. Materials and Methods

### 2.1. Participants

A total of 138 women took part in this study. They ranged in age from 19 to 47 years old. The majority of respondents were between 20 and 30 years old (*M* = 25.72; *SD* = 5.55). Among the respondents, 68 were women with diabetes, and 70 were healthy women. A larger proportion of respondents were sport practitioners, with a number of 40, while non-sport practitioners numbered 28.

Informed consent was obtained from all subjects involved in the study. The study was conducted by the third author of the study, who knew the whole protocol. All respondents were aware that personality traits, self-esteem, body-esteem, and imagery were measured. However, they were not aware of the specific purposes of the study. Written informed consent for publication has been waived due to the online form of the study. The investigation followed the ethical principles regarding human experiments as defined in the Declaration of Helsinki and received approval from the local Institutional Review Board (University of Gdańsk, 11/2015).

### 2.2. Procedure

The study took place via online questionnaires. Before testing, respondents were asked to assist in the ongoing research, which will subsequently serve as empirical material for the research paper. Respondents were assured that there were no right or wrong answers, as well as that all information contained in the questionnaires was anonymous. Respondents answered questions from 4 questionnaires. The personality traits, level of self-esteem, self-assessment of one’s own body, and ability to use imagery were examined.

### 2.3. Measurements

We used four research tools for the study.

Self-Esteem Scale (SES). To examine the level of self-esteem, we used the Self-Esteem Scale (SES) questionnaire created by Marshall Rosenberg [52] in the Polish adaptation [53]. The survey consists of 10 questions, which are answered on a 4-point scale, where 1 is “strongly agree” and 4 is “strongly disagree”. A participant can get from 10 to a maximum of 40 points. The more points they receive, the higher their self-assessment. Studies conducted on different groups confirmed that the Polish version of the method is a reliable tool (Cronbach’s alpha = 0.81–0.83) [54].

Body Esteem Scale (BES) by Franzoi and Shields [55] in the Polish adaptation [56]. This questionnaire measures respondents’ attitudes towards their own bodies. It consists of 35 items that are divided into three, gender-dependent subscales. The subscales for women are: sexual attractiveness, which measures satisfaction with the appearance of particular parts of the body associated with sexuality; weight concern, which assesses appetite and evaluates areas of the body that may be subject to possible modification through the use of an appropriate diet or increased physical activity; and physical condition, which refers to parameters related to endurance, strength, and agility. For men, there are the following subscales: physical attractiveness, upper body strength, and physical condition. The reliability of subscales of the original version was high both for females (Cronbach’s alpha from 0.80 to 0.89) and males (Cronbach’s alpha from 0.85 to 0.88). Only women took part in our study.

BFI-S Personality. The Polish version [57] of the Big Five Inventory-Short (BFI-S) [58] was used in our study. It is a fifteen-item tool with a 7-point Likert’s scale, where 1 means definitely not, and 7 means definitely yes. This questionnaire is used to measure personality within the five-factor personality theory: extraversion, openness to experience, agreeableness, neuroticism and conscientiousness. As it is a short form, this scale is more often used in exploratory research [59,60]. The reliability (Cronbach’s alpha) of the subscales in the original version are as follows: extraversion (0.62); openness to experience (0.73); agreeableness (0.50); neuroticism (0.57); and conscientiousness (0.67) [39]. The reliability (Cronbach’s alpha) of the subscales in this study are as follows: extraversion (0.65); openness to experience (0.72); agreeableness (0.56); neuroticism (0.54); and conscientiousness (0.58).

Imagination in Action. The questionnaire is a short form of the Polish questionnaire Imagination in Sport (ISQ) [61] modified for this study. It consists of 21 questions, with a 5-point Likert’s scale, where 1 means “not at all” and 5 means “completely yes”. The test consists of two parts: the first has 18 questions and refers to the specific scenarios provided in the instructions that the subjects are asked to imagine, while the second part assesses the general tendency to use imagery, i.e., general imagery—it includes 3 questions. The test consists of 7 subscales: physiological feelings (noticeable changes in body functioning), modalities (use of senses besides the visual sense), ease/control (ease and control of the imagined scene), perspective (juggling of different perspectives of the imagined scene), affirmations (positive attitude during imagery), visual (visual sense), and general (general tendency to use imagery). The higher the score on each scale, the more easily and effectively a person uses imagery. In the original version, respondents are asked to imagine the start in high-level competition. The short version of Imagination in Sport (ISQ-S) was constructed based on the best factor loadings of the original subscales. Since this is a new tool in Polish conditions, authors of ISQ-S—Budnik-Przybylska and Karasiewicz provide psychometric data of the Polish version: CFI = 0.932; TLI = 0.915; RMSEA = 0.054 (*N* = 495) and the Cronbach’s alpha statistics for each of the scales in the analyzed group: (Cronbach’s alpha): feelings (0.82), modalities (0.77); ease control (0.70), perspective (0.73), affirmation (0.81), visual (0.63), and general (0.71). In the instructions of this study, the respondents were asked to imagine doing any physical activity, i.e., riding a bike, before they started answering the questions included in the test. In the current study, each subscale showed the following reliability (Cronbach’s alpha): feelings (0.90), modalities (0.85), ease control (0.85), perspective (0.82), affirmation (0.94), visual (0.82), and general (0.76).

### 2.4. Statistical Analysis

The assumption of normality was tested using the Shapiro–Wilk test, assuming that W > 0.90 and *p* < 0.01 support the null hypothesis of normality. Due to the significant rejection of the null hypothesis in most cases, we decided to use the nonparametric Mann–Whitney procedure to test differences between groups. A *p*-value < 0.005 was declared as a significant difference between groups in the sum of ranks, and rb (rank-biserial correlation) > 0.30 was declared as a meaningful (non-trivial) effect. For the group comparisons, nonparametric Mann–Whitney U tests were used. Statistical calculations were performed using Statistica for Windows v. 10 PL.

## 3. Results

Firstly, we verified the first hypothesis, whether women diagnosed with diabetes exhibit higher levels of neuroticism and conscientiousness compared to healthy women.

Statistically significant differences were found between diabetic and healthy women (Table 1). The diabetic women scored higher on the neuroticism scale and consciousness—the last was marginally significant. In contrast, healthy women scored higher on the extraversion scale.

Subsequently, we verified the second research question: are there differences between diabetic women participating in sport and those not participating in sport in levels of personality traits, self-esteem, body evaluation, and ability to use imagery? We found only two significant differences, i.e., affirmations and sexual attractiveness, where active diabetic women obtained higher results. However, we also found that active diabetic women present a marginally significant lower level of extraversion (Table 2).

## 4. Discussion

The aim of the study is twofold: (1) to find differences in personality traits, self-esteem, body-esteem, and imagery between healthy women and women with diabetes; and (2) to verify whether there are differences in the analyzed factors among women with diabetes who participate in sports compared to those who do not.

Answering the first question: are there differences between healthy and diabetic women in the level of personality traits, self-esteem, body esteem, and ability to use imagery? We have discovered that diabetic women are characterized by differences in personality, i.e., higher neuroticism and consciousness, as well as lower extraversion. The obtained results are in line with previous research [20,24,27,28,29,30,31], which showed that people with diabetes are more conscientious and neurotic than healthy people. This implies that they possess greater organizational skills and a firm determination to reach goals, along with a strong tendency for volatile moods and a diminished capacity to manage difficult situations [32]. This may transfer to coping with the disease, disease control, and test scores. People with diabetes, driven by high levels of conscientiousness, may have a continual aspiration to achieve better test results, thereby fostering motivation to maintain controlled blood sugar levels [29].

However, with high levels of neuroticism, they may find it difficult in tough situations. A different study [62] on the personalities of individuals with diabetes showed that they have lower extraversion than healthy people and higher emotionality. Furthermore, the research conducted by Szymanska [20] indicated that diabetic people struggle more with managing stressful and difficult situations.

It has been acknowledged that a disease such as diabetes imposes a substantial responsibility on individual health management. It not only impacts but can also arise as a consequence of obesity [63]. However, delineating the precise cause and effect relationship between these factors presents a significant challenge. Based on personality traits, there are noticeable correlations between obesity and neuroticism [63]. In the study by Mukherjee et al. [30] examining personality traits in individuals with type 2 diabetes, categorized by obesity and normal weight, it was revealed that the neuroticism level was significantly higher in both the normal-weight diabetes group and the obese diabetes group as compared to the control group. Moreover, similarly to previous studies, diabetes patients presented significantly higher neuroticism than healthy controls. Admittedly, neurotic people are more prone to experiencing feelings of anxiety, anger, jealousy, sadness, or guilt. Food serves as an easy means of emotional regulation, which can lead to overuse, obesity [64], and increased morbidity [65]. Moreover, females are more likely to experience emotional eating than males [64]. However, there is a necessity to obtain more comprehensive and precise data, which should be combined with information regarding the timing of the diagnosis. This would give a broader perspective on observing the difference in management between pre-disease and post-disease individuals.

This research supports the idea that some character traits occur on a purely molecular level [66]. Both serotonin and dopamine influence how we feel. Variations in their levels among individuals contribute to the diversity of personalities observed. Dopamine is associated with curiosity and the desire to discover new things, while serotonin is linked to the inclination towards compromise, contrasting with tendencies exhibited in neurotic attitudes. Notably, serotonin and dopamine are produced during physical activity [66].

Answering the second research question, we have found that diabetic women who practice sports assess their bodies more favorably in terms of sexual attractiveness and use imagery better in terms of affirmations than healthy individuals who do not practice sports. These results are in line with previous studies in which individuals who engage in regular physical exercise possess a realistic appraisal of their physical bodies [51]. Also in a study by Ziemianek et al. [50], in which the authors examined the self-esteem and body image of women and men who exercise at the gym, the outcomes from the BES questionnaire show that physically active people rate their bodies above average. The obtained results are also in line with previous research in which athletes demonstrated enhanced confidence through the use of imagery for this objective [43]. Furthermore, imagery ability may be treated as a predictor of self-confidence [41,42].

Enhanced self-confidence among individuals engaged in physical activity, coupled with the practice of affirmations, might positively impact their subjective perception of sexual attractiveness. This could also correlate with improved mood, facilitated by serotonin release resulting from exercise [67].

The connection between feeling better and physical activity is related to serotonin. Several studies [67] have demonstrated a link between serotonin-related measures and mood within the typical range. In one study, the reduced function of serotonin receptors in platelets was connected with a decreased mood, ref. [68], whereas another study found that higher levels of serotonin in the blood were correlated with an improved mood [69].

The limitation of this study is the relatively small number of participants. Moreover, we did not indicate other variables such as educational level, socioeconomic level, care for healthy eating, or other outdoor activities that are not sports, as other variables could also be influencing. However, we obtained similar results to previous studies [28,34] on personality and revealed other unique findings. It would be highly beneficial and interesting for future studies to investigate the variations over time, from the absence of the disease, through the moment of diagnosis, followed by the management of diabetes after diagnosis. This would give interesting insights into how the disease of diabetes could affect personality traits, if such an effect exists. Additionally, future research could explore the effects of the exercise intervention on the variables being examined. Another limitation of this study was the focus on the women group only. It is worth repeating a similar study in the future on the group of men. However, the rationale behind restricting the participant pool to women in this study stems from the fact that in Poland, more women have diabetes than men [20]. The next limitation of our study is the usage of the unpublished questionnaire of the BFI-S; however, we provided the reliability of the subscales in our study.

In contrast, a strong point of this study is its focus on a group of individuals with diabetes, who are typically difficult to engage for research purposes. The study could have used a self-assessment questionnaire examining individual aspects instead of an overall self-assessment. In the future, other studies could involve asking participants about their disease status, whether they have good or bad test results, or whether their blood glucose levels are regulated or not.

## 5. Conclusions

To conclude our study, we found differences in the analyzed factors between groups. We found that women with diabetes were characterized by significantly higher neuroticism, lower extraversion, and higher conscientiousness. We also found that women with diabetes who practice sport rated their bodies more highly in terms of sexual attractiveness and made better use of imagined affirmations than women without diabetes who were not active. Our results may provide new insight into people with diabetes coping with this disease. Our findings suggest that participation in sports activities and the application of imagery constitute a natural and cost-free resource that is easy to use and does not require any expensive equipment. It helps not only with medical treatment but also with self-development.

## Figures and Tables

**Table 1 healthcare-12-00857-t001:** Comparison of diabetic and healthy women.

Variable	Diabetic Women *N* = 68		Healthy Women *N* = 70				
	*M*	*SD*	*M*	*SD*	U Mann–Whitney	*Z*	*p*
Openness	5.30	1.17	5.20	1.02	2223.50	−1.27	0.20
Conscientiousness	**5.28**	**0.87**	**5.01**	**0.89**	**1960.50**	**−1.98**	**0.05**
Extraversion	**3.74**	**1.36**	**4.27**	**1.21**	**1839.00**	**2.31**	**0.02**
Agreeableness	5.00	1.19	4.97	1.03	2291.00	−0.40	0.69
Neuroticism	**4.87**	**1.38**	**4.42**	**0.99**	**1881.50**	**−2.12**	**0.03**
Self-esteem	26.13	4.37	25.87	5.16	2327.00	−0.27	0.79
Sexual attractiveness	3.62	0.63	3.71	0.65	2183.50	0.70	0.48
Weight control	3.18	0.87	3.17	0.88	2351.00	−0.13	0.90
Physical condition	3.23	0.80	3.32	0.71	2142.00	0.74	0.46
Physiological feelings	3.02	1.31	3.12	1.10	2252.50	0.48	0.63
Modality	3.27	1.35	3.03	1.16	2090.00	−1.24	0.22
Ease/Control	3.92	0.93	3.76	0.90	2125.00	−0.60	0.55
Perspective	3.27	1.14	3.21	1.06	2258.50	−0.20	0.85
Affirmations	3.70	1.23	3.68	1.13	2308.00	−0.48	0.63
Visual	4.16	0.88	4.02	0.82	2098.00	−0.94	0.35
General	4.22	0.85	4.21	0.79	2302.50	−0.72	0.47

Significant differences were bolded.

**Table 2 healthcare-12-00857-t002:** Comparison between diabetic women participating in sports and those not participating in sports.

Variable	Diabetic Women Participating in Sports *N* = 40		Diabetic Women Not Participating in Sports *N* = 28				
	*M*	*SD*	*M*	*SD*	U Mann–Whitney	*Z*	*p*
Personality							
Openness	5.48	1.17	5.05	1.14	436.00	−1.54	0.12
Conscientiousness	5.30	0.97	5.25	0.72	535.50	−0.30	0.76
Extraversion	3.44	1.28	4.15	1.40	412.00	1.84	0.07
Agreeableness	5.14	1.16	4.80	1.22	473.50	−1.07	0.28
Neuroticism	4.71	1.27	5.10	1.51	467.50	1.15	0.25
Self-esteem							
Self-esteem	25.85	4.44	26.54	4.32	524.00	0.44	0.66
Body esteem							
Sexual attractiveness	**3.18**	**1.27**	**2.79**	**1.34**	**394.50**	**−2.06**	**0.04**
Weight control	3.42	1.36	3.07	1.35	478.00	−1.02	0.31
Physical condition	3.88	1.04	3.96	0.76	454.00	−1.31	0.19
Imagery							
Physiological feelings	3.39	1.10	3.10	1.19	458.50	−1.26	0.21
Modality	4.08	1.01	3.15	1.32	471.50	−1.10	0.27
Easiness	4.15	0.98	4.18	0.75	555.00	−0.06	0.96
Perspective	4.37	0.72	4.00	0.98	478.00	−1.02	0.31
Affirmations	**3.75**	**0.63**	**3.43**	**0.60**	**332.50**	**−2.83**	**0.00**
Visual	3.30	0.93	3.02	0.77	437.50	−1.52	0.13
General	3.36	0.82	3.04	0.74	535.50	−0.30	0.76

Significant differences were bolded.

## Data Availability

The raw data supporting the conclusions of this article will be made available by the authors, without undue reservation.

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
