# Peer review of "Why Should Diabetic Women Be Active?—The Role of Personality, Self-Esteem, Body-Esteem, and Imagery"

_healthcare, 2024, doi:10.3390/healthcare12080857_

Round 1

Reviewer 1 Report

Comments and Suggestions for Authors

In this manuscript, the authors investigate the differences in aspects such as personality traits, self-esteem, body- esteem, and imagery between healthy women or women diagnosed with diabetes and between women with diabetes who do or do not practice sports.

First of all, the authors should justify why 56% of the citations are 10 or more years old. This outdatedness affects not only the state of the art, but also the discussion of the results. That is why resorting to some new more current references is advisable.

In lines 106-107, taking into account its context in the paragraph, the reference to gender as a predictor variable for diabetes in this sentence is not understood “The strongest predictor for people with diabetes, whether or not they engaged in sport, was gender and BMI.”

As the gender perspective is applied in the manuscript from the research, it could also be considered resorting to this perspective from the language, such as, for example, using “he/she” instead of “he” as occurs in lines 134-135 “Motor imagery is important for an athlete because, by refining and transforming  them, he tries to perform the movement as he has imagined it.”

Review the sentence “Respondents with diabetes made up half of the group and there were 68 and 70 healthy women were 80” in line 171.

Regarding the analysis of the results, only the variables sex, age and level of physical activity have been considered. The sample does not indicate other variables such as educational level, socioeconomic level, care for healthy eating, or other outdoor activities that are not sports, as other variables could also be influencing. Therefore, the study and its potential could be quite limited.

Some limitations and future lines of research are included, but others could be added, such as considering more variables that are not being considered in the present study. It would also be necessary to consider the local limitation of the sample and the study compared to the generalization that the hypotheses and the research questions imply. Finally, I think that these aspects would make more sense in the conclusions section than in the Discussion section. In fact, the conclusions section is rarely shorter than the abstract.

Comments on the Quality of English Language

Review the wording in some sentences: “The results of the study, they undoubtedly confirmed the hypothesis that personality traits may play an”

“In another population-based study, Attanayake et al. [21], it was found that people with diabetes”  “Very interesting and useful in the future in similar studies would be to investigate...”

Author Response

Dear Reviewer,

please find attached the revision of our manuscript entitled “ Why should diabetes women be active?- the role of personality, self-esteem, body- esteem and imagery by D. Budnik-Przybylska, M. Fituch, A. Kowalewska.

First, we are very grateful for the opportunity to correct our original manuscript version. The additional time allowed us to review the manuscript in detail and prepare the responses to the Editor’s as well as the Reviewers' comments.  We thank all the Reviewers for the detailed reviews, valuable comments, and suggestions, which helped us significantly improve our manuscript. We applied them in the new, current version of the manuscript.

We believe that applied changes in the actual version of the manuscript increased its quality and will allow for a better understanding of our research.

We hope that it will fit within the criteria of the journal. We are very grateful in advance for including this version in further steps of the submission procedure.

The responses to the comments given by the Editor and Reviewers are listed below.

Comments and Suggestions for Authors

In this manuscript, the authors investigate the differences in aspects such as personality traits, self-esteem, body- esteem, and imagery between healthy women or women diagnosed with diabetes and between women with diabetes who do or do not practice sports.First of all, the authors should justify why 56% of the citations are 10 or more years old. This outdatedness affects not only the state of the art, but also the discussion of the results. That is why resorting to some new more current references is advisable.

Answer: Thank you for noticing that, we added many new references according to your suggestions.

In lines 106-107, taking into account its context in the paragraph, the reference to gender as a predictor variable for diabetes in this sentence is not understood “The strongest predictor for people with diabetes, whether or not they engaged in sport, was gender and BMI.”

Answer: We addressed this by indicating that it serves as a predictor of physical activity. The most influential predictor of physical activity among individuals with diabetes, regardless of participation in sports, was found to be gender and BMI. This underscores the significance of gender in determining physical activity levels, particularly when considering its implications for health.

As the gender perspective is applied in the manuscript from the research, it could also be considered resorting to this perspective from the language, such as, for example, using “he/she” instead of “he” as occurs in lines 134-135 “Motor imagery is important for an athlete because, by refining and transforming  them, he tries to perform the movement as he has imagined it.”

Answer: Thank you for this notification. We corrected that.

Review the sentence “Respondents with diabetes made up half of the group and there were 68 and 70 healthy women were 80” in line 171.

Answer:  Thank you for that comment. We corrected that sentence.

Regarding the analysis of the results, only the variables sex, age and level of physical activity have been considered. The sample does not indicate other variables such as educational level, socioeconomic level, care for healthy eating, or other outdoor activities that are not sports, as other variables could also be influencing. Therefore, the study and its potential could be quite limited.

Answer: We agree with that comment - we did not ask respondents for other variables. We added that information to the limitation.

Some limitations and future lines of research are included, but others could be added, such as considering more variables that are not being considered in the present study. It would also be necessary to consider the local limitation of the sample and the study compared to the generalization that the hypotheses and the research questions imply. Finally, I think that these aspects would make more sense in the conclusions section than in the Discussion section. In fact, the conclusions section is rarely shorter than the abstract.

Answer: Thank you for that comment, we added more information to the limitation section.

Comments on the Quality of English Language

Review the wording in some sentences: “The results of the study, they undoubtedly confirmed the hypothesis that personality traits may play an”

“In another population-based study, Attanayake et al. [21], it was found that people with diabetes”  “Very interesting and useful in the future in similar studies would be to investigate...”

Answer: We corrected the quality of the English language, thanks to the support of an academic from Australia.

Reviewer 2 Report

Comments and Suggestions for Authors

General Comments

First of all, I would like to thank you for the opportunity to review the manuscript.

The topic is interesting and relevant. However, the article needs a thorough review and improvement. English needs to be revised. The articulation between themes and presentation of the theme needs to be improved. The article lacks bibliographical support, especially in the introduction and discussion. In its current state, the discussion seems somewhat confused, disjointed and with many unfounded personal opinions and assumptions.

I leave specific comments by section and lines below.

If the authors improve all the points mentioned, the article may be considered for publication.

Introduction

Lines 25-28 – The way the sentence is written is not fluid, please rephrase and add bibilographic reference(s) to support it.

Lines 28-29 – “Physical activity is recom-28 mended especially for type 2 diabetes.” – Please support this sentence with a strong bibliographic reference. For example: (ACSM, 2021)

ACSM. (2021). Overweight and Obesity. In ACSM's Guidelines for Exercise Testing and Prescription (Eleventh edition ed., pp. 296-300). Wolters Kluwer Health.

Lines 40-42 – The authors present and frame the topic but don't always reinforce it with references. Please consider these two sentences and reinforce them with literary references.

Lines 45-46 – This sentence is confused, please try to clarify it.

Lines 47-48 – Please clarify: “is noted in children and adolescents with diabetes

Lines 54-49 – Please add references.

Lines 61-62 – The conjugation of the verb term is not correct, please check.

Line 64 – Please correct the 2nd square bracket of the reference "[13[" to "[13]".

Line 65 – This sentence “Diabetes is an incurable and chronic disease” is partially incorrect. When we refer to "diabetes", we are including all types of diabetes, including gestational diabetes, which can disappear after childbirth. Please review.

Lines 65-69 – Please add references.

Lines 77-79 – The authors refer here to one of the 5 personality traits, "neuroticism". A brief presentation of the main characteristics of the various personality traits would be very enriching for the reader (who may not know this and the other traits in depth). This information, which can be added in a previous paragraph, will help the reader to better understand the entire literature review presented in the introduction and, subsequently, the results obtained. This need is reinforced by the content presented in the following paragraphs...

Lines 106-107 – Authors mention "the strongest predictor" but of what? Please clarify.

Lines 111-116 – Please add references to all these sentences.

Lines 120-124 – Reading the introduction, I have felt that the authors "jump from topic to topic2 with short and sometimes poorly articulated paragraphs. At the top, they've already touched on one training method, Pilates. Two pages later, after mentioning psychological characteristics, they're back to talking about another training method, Hiit. The internal articulation of the introduction is fragile and deserves to be considered in order to promote a more fluid reading for the reader.

Lines 125-127 – References…

Line 136 – I understand that you used "he" because you were referring to the athlete, but both MDPI and the academic world are asking for neutral language, so I suggest using "he/she".

Lines 135-138 – References…

Lines 148-149 – “The subjects of the study were psychological variables, namely self-esteem levels, 148 self-perception of their own body, personality traits, and the ability to use imagery” - The subjects or the variables under study? Which study? This paragraph is not linked to the previous one. Please review.

Lines 152-155 – Why doesn't the study include male subjects? Please explain and justify this methodological choice, which could be seen as a weakness of the study.

Lines 163-166 – Setting hypotheses in research implies a level 3 study, in which the hypotheses are presented and supported taking into account previous literature. In this sense, I would like to ask the authors to include references that support each of the hypotheses.

 Methodology

Lines 170-171 – Please check the number of participants with diabetes, were they 68 or 70?

Lines 172-173 – Please clarify which sub-group you're referring to, I'm guessing it's women with diabetes.

Line 178 Section Procedure – Please clarify the level of blindness of the participants.

Line 191 – Add the reliability of the SES translated questionnaire.

Line 192 – Explain the interpretation results of BES scale.

Line 205 – Add the reliability of the BFIS translated questionnaire.

Line 206 – Is the ISQ questionnaire validated for Polish? Please add this information. How did the authors “adapted” the questionnaire? Please present these adaptations and justify it.

Line 215 – It is missing the “statistical analysis section”. Authors must describe all the statistical treatment they applied to obtain the results, as well as identify the level of significance adopted.

 Results

Line 223 – Table 1: please add a note with all the abbreviations. Why did the authors not bold the statistically significant result (p=0.05) of Conscientiousness?

Line 228 – Please add the comma missing after “i.e.”, which should be “i.e.,”

Discussion

Line 241 - At the beginning of the phrase "we have..." the W must be capitalized.

Line 246 – Please add a reference to support this possibility.

Lines 239-254 – The discussion for this 1st question is not very in-depth and uses few references to discuss the data found. It would be important and interesting to reinforce and deepen this discussion.

Lines 260-263 – It is imperative to reinforce with literature references.

Lines 264-269 – I apologize to the authors, but I cannot understand the insertion of this paragraph here... At no point in the manuscript was gestational diabetes addressed, nor was there any mention of the effectiveness of physical exercise programs. Therefore, the presentation of this paragraph must be reviewed and articulated with the others.

Lines 270-277 - When reading this paragraph without any reference and a hypothesis date, it seems that the authors are "trying to guess". Guessing has no place in science... Please, I ask the authors to support this paragraph with references and clarify whether they are actually guessing...

Line 290 – The statement of “Diabetes does not affect body perception, it is only affected by physical activity” is very powerful. This statement should be supported by an even more holistic assessment than the single use of a single questionnaire, and the sample does not allow for the full generalization of the results (it is a small, convenience sample, not representative of the population). Therefore, I suggest reformulating this sentence, as well as the sentence in line 262-263 to something like "For these results/participants..."

Lines 291-295 – In the paragraph above it seems that the discussion of question 2 is closed, the authors make a paragraph and discuss it again. Immediately afterwards, the next paragraph recovers the topic of serotonin that was discussed 2 paragraphs above. This alternation of themes makes reading confusing and unattractive for the reader. I ask the authors to review and reorganize this discussion.

Line 307 – I agree with the authors, but it would be important to reinforce with more references.

Lines 308-310 - The authors should better explain these weaknesses, namely the fact of the lack of representativeness of the sample. Still in relation to the sample, the inclusion of only women is also a weakness that must be presented and justified.

Line 313 - As a suggestion for a future study, it could be interesting to see the effect of the exercise intervention on the variables under study (in fact, the authors have a paragraph that was somewhat lost in the discussion on this topic).

 Conclusion

Line 324 – Please explain what you mean with this sentence.

Comments on the Quality of English Language

The article is written in a very fluid way, with some grammatical errors and bad verb conjugations. The article needs an in-depth linguistic review.

Author Response

Dear Reviewer,

please find attached the revision of our manuscript entitled “ Why should diabetes women be active?- the role of personality, self-esteem, body- esteem and imagery by D. Budnik-Przybylska, M. Fituch, A. Kowalewska.

First, we are very grateful for the opportunity to correct our original manuscript version. The additional time allowed us to review the manuscript in detail and prepare the responses to the Editor’s as well as the Reviewers' comments.  We thank all the Reviewers for the detailed reviews, valuable comments, and suggestions, which helped us significantly improve our manuscript. We applied them in the new, current version of the manuscript.

We believe that applied changes in the actual version of the manuscript increased its quality and will allow for a better understanding of our research.

We hope that it will fit within the criteria of the journal. We are very grateful in advance for including this version in further steps of the submission procedure.

The responses to the comments given by the Editor and Reviewers are listed below.

General Comments

First of all, I would like to thank you for the opportunity to review the manuscript.

The topic is interesting and relevant. However, the article needs a thorough review and improvement. English needs to be revised. The articulation between themes and presentation of the theme needs to be improved. The article lacks bibliographical support, especially in the introduction and discussion. In its current state, the discussion seems somewhat confused, disjointed and with many unfounded personal opinions and assumptions.

I leave specific comments by section and lines below.

If the authors improve all the points mentioned, the article may be considered for publication.

Answer:  Thank you for your comments and detailed review. We applied your suggestion in the new, current version of the manuscript. We believe that applied changes in the actual version of the manuscript increased its quality.

Introduction

Lines 25-28 – The way the sentence is written is not fluid, please rephrase and add bibilographic reference(s) to support it.

Answer: Thank you for your comment. We corrected the sentence and we added a new reference (Palermi et al., 2022). 

Lines 28-29 – “Physical activity is recommended especially for type 2 diabetes.” – Please support this sentence with a strong bibliographic reference. For example: (ACSM, 2021)

ACSM. (2021). Overweight and Obesity. In ACSM's Guidelines for Exercise Testing and Prescription (Eleventh edition ed., pp. 296-300). Wolters Kluwer Health.

Answer: We added that reference and added (Palermi et al., 2022).

Lines 40-42 – The authors present and frame the topic but don't always reinforce it with references. Please consider these two sentences and reinforce them with literary references.

Answer: Thank you for this notification. We reinforced it with new reference (Ouyang et al., 2020). Lines 45-46 – This sentence is confused, please try to clarify it.

Answer: We changed this sentence and we added a new reference to reinforce the statement about high self-esteem (Khampirat, 2020).

Lines 47-48 – Please clarify: “is noted in children and adolescents with diabetes

Answer: We had expanded this sentence from the wider point of view “Consequently, the child, and later the adolescent, confronts the psychological challenges of the condition, a burden not shared by their healthy peers. This experience does not escape the notice of the developing personality [20].

Lines 54-49 – Please add references.

Answer: Thank you, we have done that - Moreover, these differences and associations vary based on gender (Kokoszka et al., 2022).

Lines 61-62 – The conjugation of the verb term is not correct, please check.

Answer: Thank you, we corrected the conjugation.

Line 64 – Please correct the 2nd square bracket of the reference "[13[" to "[13]".

Answer: We corrected that.

Line 65 – This sentence “Diabetes is an incurable and chronic disease” is partially incorrect. When we refer to "diabetes", we are including all types of diabetes, including gestational diabetes, which can disappear after childbirth. Please review.

Answer: Thank you, we corrected that.

Lines 65-69 – Please add references.

Answer: The sentence was expanded and we add reference.

Glucose concentration values are influenced by a range of physiological and metabolic factors, including physical activity and acute psychological stress, alongside the effects of meals and insulin (Sevil, 2021).

Lines 77-79 – The authors refer here to one of the 5 personality traits, "neuroticism". A brief presentation of the main characteristics of the various personality traits would be very enriching for the reader (who may not know this and the other traits in depth). This information, which can be added in a previous paragraph, will help the reader to better understand the entire literature review presented in the introduction and, subsequently, the results obtained. This need is reinforced by the content presented in the following paragraphs…

Answer: We expanded theory of the Big 5 by (McCrae and Costa, 1999).

Lines 106-107 – Authors mention "the strongest predictor" but of what? Please clarify.

Answer: Of physical activity. We corrected that.

Lines 111-116 – Please add references to all these sentences.

Answer: References were added: (An et al., 2020),  (Lekue et al. 2022).

Lines 120-124 – Reading the introduction, I have felt that the authors "jump from topic to topic2 with short and sometimes poorly articulated paragraphs. At the top, they've already touched on one training method, Pilates. Two pages later, after mentioning psychological characteristics, they're back to talking about another training method, Hiit. The internal articulation of the introduction is fragile and deserves to be considered in order to promote a more fluid reading for the reader.

Answer: This is why we decided to cancel this argument and focus on the main topic.

Lines 125-127 – References…

Answer: We developed this subject by changing the whole sentence and adding reference: “Studies show that people feel better after doing physical activity (An et al., 2020). Happiness hormones such as dopamine, serotonin, and endorphin are produced during and after physical activity (Lekue et al. 2022). Which has a soothing effect on the nervous system.”

Line 136 – I understand that you used "he" because you were referring to the athlete, but both MDPI and the academic world are asking for neutral language, so I suggest using "he/she".

Answer: Thank you. We corrected that.

Lines 135-138 – References…

Answer: We corrected that and putting new explanation.

Lines 148-149 – “The subjects of the study were psychological variables, namely self-esteem levels, 148 self-perception of their own body, personality traits, and the ability to use imagery” - The subjects or the variables under study? Which study? This paragraph is not linked to the previous one. Please review.

Answer: We deleted that part.

Lines 152-155 – Why doesn't the study include male subjects? Please explain and justify this methodological choice, which could be seen as a weakness of the study.

Answer: We wanted to focus on women because according to the study of Topor-Madry et al, 201  more women have diabetes and they become less active (Mayo et al. 2019, Tsai et al. 2014). We added that information and suggested a repetition of this study on men in future research.

Lines 163-166 – Setting hypotheses in research implies a level 3 study, in which the hypotheses are presented and supported taking into account previous literature. In this sense, I would like to ask the authors to include references that support each of the hypotheses.

Answer: Thank you for noticing, we had explained that. “According to [34] neuroticism is associated with a diminished subjective sense of diabetic management and heightened levels of anxiety, particularly among women with diabetes mellitus (DM). Enhancing self-efficacy in diabetes management could potentially aid individuals in coping with anxiety symptoms, especially among those exhibiting traits of neuroticism”.

A study by Ziemianek et al. [50], in which the authors examined the self-esteem and body image of women and men who exercise at the gym, showed that physically active people rate their bodies above average. In a study by Budnik-Przybylska et al. [51], dancers showed higher imagery abilities as well as better ratings of their bodies in terms of physical fitness.

 Methodology

Lines 170-171 – Please check the number of participants with diabetes, were they 68 or 70? 

Answer: Thank you for your comments. We corrected our mistake.

Lines 172-173 – Please clarify which sub-group you're referring to, I'm guessing it's women with diabetes.

Answer: We clarified that statement.

Line 178 Section Procedure – Please clarify the level of blindness of the participants.

Answer: We added: Informed consent was obtained from all subjects involved in the study. Written informed consent for publication has been waived due to the online form of the study.

Line 191 – Add the reliability of the SES translated questionnaire.

Answer: We added the following statement: Studies conducted on different groups confirmed that the Polish version of the method is a reliable tool (& Cronbach's = 0.81-0.83) (Łaguna et al. 2007).

Line 192 – Explain the interpretation results of BES scale.

Answer: Thank you. We added the interpretation results of BES scale

Line 205 – Add the reliability of the BFIS translated questionnaire.

Answer: We added the reliability of each subscale from the original version. 

Line 206 – Is the ISQ questionnaire validated for Polish? Please add this information. How did the authors “adapted” the questionnaire? Please present these adaptations and justify it.

Answer: We added the required information:

Line 215 – It is missing the “statistical analysis section”. Authors must describe all the statistical treatment they applied to obtain the results, as well as identify the level of significance adopted.

Answer:  Thank you. We added  Statistical Analysis section.

 Results

Line 223 – Table 1: please add a note with all the abbreviations. Why did the authors not bold the statistically significant result (p=0.05) of Conscientiousness?

Answer: We bolded the results of conscientiousness.

Line 228 – Please add the comma missing after “i.e.”, which should be “i.e.,”

Answer:  A coma was added.

Discussion

Answer: We reorganized the discussion section to be more consistent.

Line 241 - At the beginning of the phrase "we have..." the W must be capitalized.-

Answer: we capitalized “W”

Line 246 – Please add a reference to support this possibility.

Answer: We put relevant references and reorganized this section.

Lines 239-254 – The discussion for this 1st question is not very in-depth and uses few references to discuss the data found. It would be important and interesting to reinforce and deepen this discussion.

Answer: We cleaned part of the discussion by linking paragraphs concerning answering the first question. We also added more references.

Lines 260-263 – It is imperative to reinforce with literature references.

Answer: We reorganised that part.

Lines 264-269 – I apologize to the authors, but I cannot understand the insertion of this paragraph here... At no point in the manuscript was gestational diabetes addressed, nor was there any mention of the effectiveness of physical exercise programs. Therefore, the presentation of this paragraph must be reviewed and articulated with the others.

Answer: We deleted that part.

Lines 270-277 - When reading this paragraph without any reference and a hypothesis date, it seems that the authors are "trying to guess". Guessing has no place in science... Please, I ask the authors to support this paragraph with references and clarify whether they are actually guessing…

Answer: We reorganised that part and added more references.

Line 290 – The statement of “Diabetes does not affect body perception, it is only affected by physical activity” is very powerful. This statement should be supported by an even more holistic assessment than the single use of a single questionnaire, and the sample does not allow for the full generalization of the results (it is a small, convenience sample, not representative of the population). Therefore, I suggest reformulating this sentence, as well as the sentence in line 262-263 to something like "For these results/participants..."

Answer: Thank you for that comment, we reorganized the discussion section.

Lines 291-295 – In the paragraph above it seems that the discussion of question 2 is closed, the authors make a paragraph and discuss it again. Immediately afterwards, the next paragraph recovers the topic of serotonin that was discussed 2 paragraphs above. This alternation of themes makes reading confusing and unattractive for the reader. I ask the authors to review and reorganize this discussion.

Answer: Thank you for that comment, as we mentioned before we reorganized the discussion section.

Line 307 – I agree with the authors, but it would be important to reinforce with more references.

Answer: We included relevant references.

Lines 308-310 - The authors should better explain these weaknesses, namely the fact of the lack of representativeness of the sample. Still in relation to the sample, the inclusion of only women is also a weakness that must be presented and justified.

Answer:We added information to that part.

Line 313 - As a suggestion for a future study, it could be interesting to see the effect of the exercise intervention on the variables under study (in fact, the authors have a paragraph that was somewhat lost in the discussion on this topic).

Answer: Thank you for that comment, we included your suggestion.

 Conclusion

Line 324 – Please explain what you mean with this sentence.

Answer: We reformulated that section.

Comments on the Quality of English Language

The article is written in a very fluid way, with some grammatical errors and bad verb conjugations. The article needs an in-depth linguistic review.

Answer: We corrected the quality of  English language.

Round 2

Reviewer 2 Report

Comments and Suggestions for Authors

2nd Review

General Comments

I do like to start by congratulating the authors on their work on the review. The manuscript has improved significantly, the authors have done a good job of reinforcing it with appropriate literature and making it more fluid to readers.

Nevertheless, the manuscript still needs to be revised. The use of a non-valid scale weakens the methodological quality of the study, so this issue needs to be made clear and discussed in more detail.

Best wishes to the authors.

Specific Comments

Line 3 – Please delete the extra space in “body- esteem”.

Line 4 – Please remove the extra space between the author’s name and their affiliations.

Line 7 – Please add a space between the “;” and the author’s e-mail.

Introduction

Line 54 – Please remove the extra space following reference 16.

Lines 61-62 – Please introduce the reference “(Khampirat, 2020)” according the to the journal’s norms.

Line 85 – In the first revision the authors claimed that “Diabetes is an incurable and chronic disease”. After my comment arguing that the sentence was partially incorrect, due to when we refer to "diabetes", we are including all types of diabetes, including gestational diabetes, which can disappear after childbirth. The authors retrieved the word “incurable”. Nevertheless, disease tends to be chronic, but if it is treated (including the diabetes triad of changing to a healthier lifestyle, practicing physical activity and eating better) it can in some cases be reversed; gestational diabetes can also disappear after the baby is born. Having said that, I would again advise the authors to revise the phrase, I believe that "diabetes is a disease that tends to be chronic" would be a more correct statement.

Line 176 – According to whom? Please introduce the author’s name and maintain the reference according to the journal’s guidelines.

Methodology

Line 202 Section Procedure – Please clarify the level of blindness of the participant, i.e. did the participants or the researchers who applied the protocols know the purpose of the study? If not, the study is double-blind, if at least one of these groups knew, it is single-blind.

Lines 230-237 – The BFIS scale has not been validated for the language used. The reference introduced is not published and, considering that the authors mention the reliability values of the original scale, it follows that in this non-validation work, reliability has not been tested. The use of a non-valid scale represents a considerable internal weakness in the study, which may not be acceptable for publication in a journal of the quality of Healthcare. The authors should acknowledge this weakness, defend it and report it in the limitations section of the discussion.

Lines 242-243 – As requested, the authors justified the adaptation made to the ISQ questionnaire, but I'm still not sure that this adaptation was safe. Please add more details, who reviewed or decided on this change? Were there meetings and discussions with experts, was the short version of the questionnaire tested beforehand?

Lines 258-260 - Remove the advance from this paragraph. The normality of the sample has been tested, I suppose, please add that information, as well as the level of significance adopted.

Discussion

Line 291 – Please add the comma missing after “i.e.”, which should be “i.e.,”. Review this question in all text.

Lines 343-348 – Remove the advance from this paragraph.

Lines 352-353 – The authors mentioned “previous studies”, so they should cite them.

Conclusion – Much clearer, great reorganization.

Author Response

General Comments

I do like to start by congratulating the authors on their work on the review. The manuscript has improved significantly, the authors have done a good job of reinforcing it with appropriate literature and making it more fluid to readers.

Nevertheless, the manuscript still needs to be revised. The use of a non-valid scale weakens the methodological quality of the study, so this issue needs to be made clear and discussed in more detail.

Best wishes to the authors.

Answer: We thank the Reviewer for the comments and the reinforcement. We applied your suggestion in the new, current version of the manuscript.

Specific Comments

Line 3 – Please delete the extra space in “body- esteem”.

Answer: We deleted the extra space.

Line 4 – Please remove the extra space between the author’s name and their affiliations.

Answer: We removed the extra space.

Line 7 – Please add a space between the “;” and the author’s e-mail.

Answer: We added a space.

Introduction

Line 54 – Please remove the extra space following reference 16.

Answer: The extra space was deleted.

Lines 61-62 – Please introduce the reference “(Khampirat, 2020)” according the to the journal’s norms.

Answer: We deleted (Khampirat, 2020)” and left only the number.

Line 85 – In the first revision the authors claimed that “Diabetes is an incurable and chronic disease”. After my comment arguing that the sentence was partially incorrect, due to when we refer to "diabetes", we are including all types of diabetes, including gestational diabetes, which can disappear after childbirth. The authors retrieved the word “incurable”. Nevertheless, disease tends to be chronic, but if it is treated (including the diabetes triad of changing to a healthier lifestyle, practicing physical activity and eating better) it can in some cases be reversed; gestational diabetes can also disappear after the baby is born. Having said that, I would again advise the authors to revise the phrase, I believe that "diabetes is a disease that tends to be chronic" would be a more correct statement.

Answer: We changed it:  Diabetes is a disease that tends to be chronic.

Line 176 – According to whom? Please introduce the author’s name and maintain the reference according to the journal’s guidelines.

We changed it: According to Woon et al. [34]

Methodology

Line 202 Section Procedure – Please clarify the level of blindness of the participant, i.e. did the participants or the researchers who applied the protocols know the purpose of the study? If not, the study is double-blind, if at least one of these groups knew, it is single-blind.

Answer: All respondents were aware that the personality traits, self-esteem, body esteem and imagery were measured. However, they were not aware of the specific purposes of the study.

Lines 230-237 – The BFIS scale has not been validated for the language used. The reference introduced is not published and, considering that the authors mention the reliability values of the original scale, it follows that in this non-validation work, reliability has not been tested. The use of a non-valid scale represents a considerable internal weakness in the study, which may not be acceptable for publication in a journal of the quality of Healthcare. The authors should acknowledge this weakness, defend it and report it in the limitations section of the discussion.

Answer: The BFIS scale was used in previous studies conducted by the first author (i. e. Budnik-Przybylska, D., Kaźmierczak, M., Przybylski, J., & Bertollo, M. (2019). Can personality factors and body esteem predict imagery ability in dancers?. Sports, 7(6), 131), MDPI) and the reliability was introduced. Now the test is being validated by the first author to Polish conditions on the group of athletes. We reported that in the limitation section. “The next limitation of our study is the usage of the unpublished questionnaire of the BFI-S, however, we provided the reliability of the subscales in our study.”

We also calculated the reliability of each scale in our study and added:

The reliability (Cronbach’s alpha) of the subscales in this study are as follows: extraversion (.65); openness to experience (.72);  agreeableness (.56);  neuroticism (.54); and  conscientiousness (.58).

Lines 242-243 – As requested, the authors justified the adaptation made to the ISQ questionnaire, but I'm still not sure that this adaptation was safe. Please add more details, who reviewed or decided on this change? Were there meetings and discussions with experts, was the short version of the questionnaire tested beforehand?

Answer: Thank you for that comment: We added: The short version of Imagination in sport (ISQ-S) was constructed based on the best factor loadings of the original subscales. Due to the fact that this is a new tool in Polish conditions, authors of ISQ-S - Budnik-Przybylska and Karasiewicz provide data psychometric data of the Polish version:  CFI = .932; TLI = .915; RMSEA = .054 (N = 495) and the Cronbach's alpha statistics for each of the scales in the analyzed group: (Cronbach’s alpha): feelings (.82), modalities (.77); ease control (.70), perspective (.73), affirmation (.81), visual (0.63), and general (.71).

Lines 258-260 - Remove the advance from this paragraph. The normality of the sample has been tested, I suppose, please add that information, as well as the level of significance adopted.

Answer: We remove the first sentence and add this part: The assumption of normality was tested using the Shapiro-Wilk test, assuming that W > 0,90 and p < 0,01 supports the null hypothesis of normality. Due to significant rejection of the null hypothesis in most cases we decided to use nonparametric Mann-Whitney procedure to test differences between groups. P-value < 0,005 was declared as a significant difference between groups in the sum of ranks and rb (rank-biserial correlation) > 0,30 as a meaningful (non trivial) effect.

Discussion

Line 291 – Please add the comma missing after “i.e.”, which should be “i.e.,”. Review this question in all text.

Answer: Thank you for noticing this.

Lines 343-348 – Remove the advance from this paragraph.

Answer: Thank you for noticing this.

Lines 352-353 – The authors mentioned “previous studies”, so they should cite them.

 Answer: Thank you that comment. We corrected that and we cited research by Woon et al., [34]. and Wheeler, K et al., (2012) [28].

Conclusion – Much clearer, great reorganization.

Answer: Thank you.